# An Analysis of the Intestinal Microbiome Combined with Metabolomics to Explore the Mechanism of How Jasmine Tea Improves Depression in CUMS-Treated Rats

**DOI:** 10.3390/foods13162636

**Published:** 2024-08-22

**Authors:** Yangbo Zhang, Yong Lin, Yifan Xiong, Jianan Huang, Zhonghua Liu

**Affiliations:** 1School of Pharmacy, Shaoyang University, Shaoyang 422000, China; zhangyblucky@163.com; 2Key Laboratory of Tea Science of Ministry of Education, Hunan Agricultural University, Changsha 410128, China; yong-lin@hunau.edu.cn (Y.L.); xyf951118@163.com (Y.X.); jianan-huang@hotmail.com (J.H.); 3National Research Center of Engineering and Technology for Utilization of Botanical Functional Ingredients, Hunan Agricultural University, Changsha 410128, China; 4Co-Innovation Center of Education Ministry for Utilization of Botanical Functional Ingredients, Hunan Agricultural University, Changsha 410128, China

**Keywords:** jasmine tea, CUMS, gut microbiome, metabolome

## Abstract

Recently, research has confirmed that jasmine tea may help improve the depressive symptoms that are associated with psychiatric disorders. Our team previously found that jasmine tea improved the depressive-like behavior that is induced by chronic unpredictable mild stress (CUMS) in Sprague Dawley (SD) rats. We hypothesized that the metabolic disorder component of depression may be related to the gut microbiota, which may be reflected in the metabolome in plasma. The influence of jasmine tea on gut microbiota composition and the association with depressive-related indexes were explored. Furthermore, the metabolites in plasma that are related to the gut microbiota were identified. SD rats were treated with control or CUMS and administrated jasmine tea for 8 weeks. The 16S rRNA gene amplicon sequencing was used to analyze the gut microbiota in feces samples, and untargeted metabolomics was used to analyze the metabolites in plasma. The results found that jasmine tea significantly ameliorated the depressive behavior induced by CUMS, significantly improved the neurotransmitter concentration (BDNF and 5-HT), and decreased the pro-inflammation levels (TNF-α and NF-κB). The intervention of jasmine tea also alleviated the dysbiosis caused by CUMS; increased the relative abundance of *Bacteroides, Blautia, Clostridium,* and *Lactobacillus*; and decreased *Ruminococcus* and *Butyrivibrio* in the CUMS-treated rats. Furthermore, the serum metabolites of the CUMS-treated rats were reversed after the jasmine tea intervention, i.e., 22 were up-regulated and 18 were down-regulated, which may have a close relationship with glycerophospholipid metabolism pathways, glycine serine and threonine metabolism pathways, and nicotinate and nicotinamide metabolism pathways. Finally, there were 30 genera of gut microbiota related to the depressive-related indexes, and 30 metabolites in the plasma had a strong predictive ability for depressive behavior. Potentially, our research implies that the intervention of jasmine tea can ameliorate the depression induced by CUMS via controlling the gut flora and the host’s metabolism, which is an innovative approach for the prevention and management of depression.

## 1. Introduction

In recent years, more and more people have been suffering from depression worldwide, causing a serious burden to the world. The common symptoms of depression involve a lack of interest, feelings of sadness, cognitive challenges, sluggish thinking, changes in eating and sleeping patterns, and other physical manifestations [1]. Dysbiosis has been linked to the development of many psychiatric and neurodegenerative diseases, including Alzheimer’s disease, depression, and schizophrenia [2], which are detrimental to human health. Studies showed that dysbiosis was considered to be a common mechanism of depression [3,4,5]. There is constant bidirectional communication between the brain and the gut microbiota via neural, hormonal, and immune interactions, which is denominated as the microbiota–gut–brain axis [6,7,8]. Disturbances of intestinal biodiversity are caused by intestinal permeability and systemic inflammation, which are related to the central nervous system [9]. The health of the gut microbiome is connected to the synthesis of neurotransmitters, myelination of neurons in the prefrontal cortex, and the development of the amygdala and hippocampus, which all impact brain function and behavior through the microbiota–gut–brain axis [10]. Anxiety and depression influence the composition of the colon microbiome, suggesting that these changes are caused by the disturbance of the microbial habitat by the activation of the stress response and altered colon motility [6].

The gut microbiota comprises the dominant microorganisms that can inhabit the human body and influence the host’s nutrition and metabolism [11]. Dysregulation of the gut microbiota may mediate depression-like behavior by affecting the host’s metabolism [12]. Metabolomics is an effective method used to analyze metabolites in the body [13]. Research has shown that the concentrations of amino acids, organic acids, sugars, and fatty acids are higher in plasma samples after CUMS treatment when compared to the control samples, suggesting that depression may be associated with the disturbance of energy metabolism, amino acid metabolism, and glucose metabolism [14]. Glutamic acid is considered to be the main excitatory neurotransmitter in the nervous system [15]. In a study, the content of glutamic acid, aspartic acid, amino malonic acid, dehydroascorbic acid, beta-alanine, glycine, and ethanolamine were reduced, while fructose, glycolic acid, erythronic acid, beta-tocopherol, 5-hydroxytryptamine, adenosine-5-monophosphate, 4,5-dimethyl-2,6-creatinine, dihydroxypyrimidine, alpha-tocopherol, myoinositol, and tetra decanoic acid were increased after CUMS intervention when compared with the healthy group, which indicated that altered energy metabolism, oxidative metabolism, and neurotransmitter disorder might be related to the pathogenesis of major depression disorder (MDD) [15]. In another study, the tyrosine produced by phenylalanine hydroxylation significantly decreased in the depression group, which had a close relationship with the lower phenylalanine hydroxylase activity [16]. In a further study, rats subjected to CUMS showed disturbances in 20 metabolites in plasma and 16 metabolites in urine compared to the control group, which were related to sphingolipid metabolism, the citrate cycle, glycolysis, fatty acid metabolism, phenylalanine metabolism, etc. [17]. The gut flora can impact the health of the host by converting indigestible carbohydrates into short-chain fatty acids (SCFAs), altering bile acids, combating pathogenic bacteria, and regulating the host’s innate and adaptive immune systems, which might be involved in pathways, such as oxidative stress metabolism, energy metabolism, and neurotransmitter metabolism [18].

Jasmine (*Jasminum*) is loved by everyone because of its unique, elegant appearance; aroma; and physiological activity, including antioxidant, antibacterial, antidiabetic, and anti-inflammatory properties [19,20,21,22]. In China, jasmine is often used to make jasmine tea [23]. Chinese jasmine tea is a type of flower tea, which is made by repeatedly mixing the aromatic flowers of *Jasminum sambac* with the tea dhool, transferring the aromatic compounds in the fresh flowers to the tea dhool [24]. Jasmine tea is popular among the public due to its elegant aroma and taste and can activate the parasympathetic nerve [25].

Jasmine tea can help people with depression by alleviating depression symptoms. However, there is still a lack of understanding regarding how jasmine tea may prevent the depression caused by CUMS. Here, CUMS was used as the animal model of depression. This study aimed to analyze whether and how the microbial composition and function modulate the metabolism of the CUMS-treated rats after jasmine tea administration. The depression-prevention effects of jasmine tea were analyzed through a variety of classic behavioral tests. After the experiment was finished, 16S rRNA sequencing and non-targeted metabolomics were performed to analyze the latent depressive-associated gut flora and metabolism. Next, the relationships between neurotransmitters, inflammation, and intestinal microbiota were investigated; meanwhile, the correlation between the intestinal microbiota and metabolites was investigated. Furthermore, the crucial metabolic pathway related to ameliorating the depression symptoms induced by CUMS after jasmine tea administration was revealed via an enrichment analysis.

## 2. Materials and Methods

### 2.1. Animals

This study involved male Sprague Dawley rats obtained from Silaikejingda Experimental Animal Co., Ltd. (Changsha, China) at 4 weeks old and kept in a specific pathogen-free facility at Hunan Agricultural University. All procedures and animal care in this research were strictly carried out in accordance with the Institutional Review Board guidelines and ethical standards set by the Animal Protection Committee of Hunan Agricultural University (Changsha, China). The rats were given one week to acclimate before the experiments began. Rats were housed at a room temperature of 25 ± 1 °C, a relative humidity of 50–60%, with a 12 h light/dark cycle. The experimental food was a common basic feed provided by Nantong Trophy Feed Company (Nantong, China), which contained carbohydrates, protein, and a small amount of fat, as well as various essential nutrients for rat growth. After adaption, rats were divided into three groups at random, including control (treated with water), model (CUMS treated with water), and NF (CUMS-treated rats treated with jasmine tea at a concentration of 64.8 mg/kg) (*n* = 10 for per group). The model and NF group experienced a range of stressors during the whole experiment (8 consecutive weeks) (Figure 1). Jasmine tea was manufactured with jasmine flowers and green tea by ourselves using a new scenting process. During the stress intervention, jasmine tea soup was prepared with drinking water and taken orally once a day at a concentration of 64.8 mg/kg (1 mL/kg/day) based on our previous research [26] until the end of the experiment. The oral dose was calculated according to the body weight of each individual rat. The Con (control) and Mod (model, which was treated with CUMS) groups received the drinking water based on the body weight of each individual rat.

### 2.2. Induction of Depressive Rat Model

The CUMS steps were performed by Sun et al. [27] with a minor adjustment. In brief, the CUMS protocol encompassed a range of mild stressors, including tail pinching, 24 h food and water deprivation, 24 h water deprivation, cold water swimming, cage tilting, dirty bedding, reversed light–dark cycle, foot shock, and physical restraint. Give one stressor (random or order) every day for 8 weeks. The workflow is displayed in Figure 1: the experiment included a stress procedure from day 1 to day 56, followed by a sugar preference test (SPT) on day 57, an open-field test (OFT) on day 58, and a forced swimming test (SFT) on day 59. The rats were kept in groups of five per cage throughout the study. Before each behavior test, the rats were acclimatized in the laboratory room for a minimum of 1 h. After each test, the rats were returned to their original cage. The control group of rats was housed in a separate room to prevent any contact with the stressed rats or any stimulation.

### 2.3. Body Weight Measurement

The CUMS treated rats were given a normal diet, except for water fasting. Dietary intake and body weight were recorded weekly during the whole experiment.

### 2.4. Behavioral Tests

The behavioral tests were performed by knowledgeable and skilled observers who were unaware of the animals’ grouping. The FST was conducted as described by S et al. [27]. The OFT was performed as described by Mou et al. [28]. The SPT was performed as described by Mou Z et al. The sugar water preference of each group was measured on day 0 (end of adaptive feeding), day 7, day 14, day 21, day 28, day 35, day 42, day 49, and day 57.

Preference for sugar water (%) = consumption of sucrose water/(total amount of sucrose water + distilled water) × 100%.

### 2.5. Fecal Sample Collection

At 8 weeks post-treatment, the fresh fecal samples of rats in each group were collected and promptly frozen in liquid nitrogen before being stored at −80 °C for subsequent analysis.

### 2.6. Tissue Collection

Following the completion of all behavioral tests, rats were anesthetized using pentobarbital sodium (Beijing Luyuan Bode Biotechnology Co., LTD, Beijing, China) injection and sacrificed after fasting for 12 h. The serum, brain, and colon tissue were excised for further experiments. The brain and colon were preserved using 4% paraformaldehyde solution (Wuhan Pinofei Biological Technology Co., LTD, Wuhan, China), the brain was examined using HE and nissl analysis, and colon was examined using HE analysis. The cerebral cortex and hippocampus, which had been separated from the brain tissues for biochemical analysis, were immediately frozen in liquid nitrogen for analysis.

### 2.7. Enzyme-Linked Immunosorbent Assay (ELISA)

The concentrations of 5-HT, BDNF, TNF-α, and NF-κB in the cerebral cortex, hippocampus, colon, and serum were measured using rat 5-HT, BDNF, TNF-α, and NF-κB ELISA kits (Cusabio Biotech., Ltd., Wuhan, China). The concentrations of 5-HT, BDNF, TNF-α, and NF-κB were assessed by measuring the absorbance with ABI Quant Studio 3 (Thermo Scientific, Wilmington, NC, USA) and creating a standard curve.

### 2.8. Hematoxylin and Eosin (HE) and Nissl Staining

The histological analysis was conducted according to Liu’s procedure [29]. Brain and colon tissue specimens were fixed in 10% formalin (Wuhan Pinofei Biological Technology Co., LTD, Wuhan, China), followed by decalcification and dehydration to achieve transparency. Subsequently, they were immersed and embedded in paraffin. Thin tissue samples measuring 3–6 μm were then obtained using a microtome (Shanghai Leica Instrument Co., LTD, Shanghai, China). These samples were dewaxed with xylene (Sinopharm Group Chemical reagent Co., LTD, Shanghai, China), treated with an aqueous ethanol series (Sinopharm Group Chemical reagent Co., LTD, Shanghai, China), and stained with H&E and Nissl for brain examinations, HE for the colon examinations, and observed using 250 pieces white light scanner (3DHISTECH, Budapest, Hungary).

### 2.9. 16S rRNA Sequencing

#### 2.9.1. Intestinal Microbial DNA Extraction

The microbial DNA was separated from fecal samples using the E.Z.N.A.^®^ soil DNA Kit (Omega Bio-Tek, Norcross, GA, USA) following the manufacturer’s instructions. The concentration and purity of the extracted DNA were assessed using a Nano Drop 2000 UV-vis spectrophotometer (Thermo Scientific, Wilmington, NC, USA), and the quality of DNA was evaluated through 1% agarose gel electrophoresis.

#### 2.9.2. 16S rRNA Sequencing and Analysis

The 16S rRNA sequencing and analysis was carried out according to Zhang et al. [26]. The V3-V4 hypervariable regions of the 16S rRNA gene in bacteria were amplified using 338F (5′-ACTCCTACGGGAGGCAGCAG-3′) and 806R (5′-GGTACHVGGG TWTCT-AAT-3′) primers through the thermocycler PCR system (GeneAmp 9700, ABI, Los Angeles, CA, USA).

Raw FASTQ files were demultiplexed using the QIIME2 demux plugin (version 2022.2) according to the unique barcodes. The data were filtered according to relevant requirements. The presence of chimeric sequences was detected and eliminated using UCHIME (version 7.1 http://drive5.com/uchime, accessed on 10 June 2022). Operational taxonomic units (OTUs) were grouped together based on a 97% similarity cutoff using UPARSE (version 7.1 http://drive5.com/uparse/, accessed on 10 June 2022). Alpha diversity metrics at the feature level, such as the observed and Chao1 richness estimator and the Shannon and Simpson diversity indices, were computed to assess the microbial diversity in individual samples. Beta diversity metrics, including Bray–Curtis, unweighted UniFrac, and weighted UniFrac distances, were employed to examine the structural differences among microbial communities among three groups (*n* = 6 per group) and then visualized via principal component analysis (PCA), which was finished with the Wekemo Bioincloud technology (https://www.bioincloud.tech, accessed on 10 June 2022) [30]. The main bacterial groups that distinguished among the three categories were determined using linear discriminant analysis effect size (LEfSe) analysis, which evaluated the impact of each genus’s abundance (LDA > 2.0 and *p* value < 0.05), which was finished with the Wekemo Bioincloud technology (https://www.bioincloud.tech, accessed on 10 June 2022) [30]. Microbial species’ relative abundances at various taxonomic levels were calculated by utilizing the R package (version R-3.4.3) called “vegan”.

### 2.10. Serum Metabolomics Analysis

The metabolite extraction of serum was mainly performed as described by the previously reported methods [31]. In short, the 100 mL samples were mixed with prechilled 80% methanol. They were then left to incubate on ice for 5 min and were centrifuged at 15,000× *g*, 4 °C for 20 min. Some of the resulting supernatants were then diluted to a final concentration containing 53% methanol using LC-MS grade water. The samples were then moved to a new Eppendorf tube and centrifuged again at 15,000× *g*, 4 °C for 20 min. Finally, the supernatant was injected into the UHPLC-MS/MS (Thermo Fisher, Karlsruhe, Germany) system for analysis [32,33].

The UHPLC-MS/MS analyses were carried out using a Vanquish UHPLC system from Thermo Fisher, Karlsruhe, Germany, connected to an Orbitrap Q Exactive^TM^ HF mass spectrometer, also from Thermo Fisher, Karlsruhe, Germany. To begin the analysis, samples were loaded onto a Hypesil Gold column (100 × 2.1 mm,1.9 μm, Waters, Milford, MA, USA) with a 17 min linear gradient flow at 0.2 mL/min. Eluents consisted of eluent A (0.1% FA in Water) and eluent B (Methanol). The solvent gradient program included the following steps: starting at 2% B for 1.5 min, increasing to 100% B over 12.0 min, maintaining 100% B for 14.0 min, decreasing to 2% B at 14.1 min, and finally ending at 2% B at 17 min. The Q Exactive^TM^ HF mass spectrometer was operated with a spray voltage of 3.2 kV, a capillary temperature of 320 °C, sheath gas flow rate set at 40 arb, and an aux gas flow rate of 10 arb.

### 2.11. Data Processing and Metabolite Identification

The UHPLC-MS/MS-generated raw data files were processed using Compound Discoverer 3.1 (CD3.1, Thermo Fisher, Wilmington, NC, USA) to conduct peak alignment, peak picking, and quantitation for each metabolite. The normalized data were then utilized to predict the molecular formula by analyzing additive ions, molecular ion peaks, and fragment ions. Subsequently, the peaks were compared with the mzCloud (https://www.mzcloud.org/, accessed on 15 July 2022), mzVault, and MassList databases to retrieve precise qualitative and relative quantitative outcomes. Statistical analyses were carried out using the software packages R (version R-3.4.3), Python (version 2.7.6), and CentOS (release 6.6). In cases where the data deviated from a normal distribution, efforts were made to normalize the data using the area-normalization method. In order to visually differentiate among the three groups (*n* = 6 per group), Orthogonal Partial Least Squares Discriminant Analysis (OPLS-DA) was employed. The effectiveness of the PLS-DA model was assessed through Hotelling’s T2 test, while the LAD score was scrutinized using Kruskal–Wallis and Wilcoxon tests. By examining the loadings in Principal Component Analysis (PCA), the specific metabolites that played a key role in distinguishing among the three groups were pinpointed (identified by a variable importance plot (VIP) score > 1.0 and *p*-values < 0.05). An analysis of pathways was performed using the Kyoto Encyclopedia of Genes and Genomes (KEGG) pathway database with the assistance of MetaboAnalyst 4.0. The overrepresentation analysis (ORA) was evaluated to discern the biological properties of the differential metabolites. Those analyses were finished with the Wekemo Bioincloud technology (https://www.bioincloud.tech, accessed on 15 July 2022) [30].

### 2.12. Data Correlation Analysis

The microbial network was evaluated using Spearman’s rank correlation analysis based on the specified conditions (Correlation > 0.1 and *p*-value < 0.05). Gut microbiota; the body weight; and the concentrations of 5-HT, BDNF, TNF-α, and NF-κB in the colon of depression rats induced by CUMS were determined by Spearman correlation analysis. A hypergeometric test was utilized for the differential metabolite enrichment analysis. Spearman correlation was used for the microbiome and metabolome correlation analysis, with only values of CC > 0.8 and *p*-value < 0.05 considered. Any *p*-values < 0.05 were deemed statistically significant.

### 2.13. Statistical Analysis

The statistical analysis was performed using SPSS version 23.0 (SPSS, Chicago, IL, USA). Continuous variables, including sucrose preference, numbers in OFT, immobility time in FST, and body weight, were analyzed using one-way analysis of variance (ANOVA). The results were analyzed as mean ± standard error of the mean (SEM) unless stated otherwise. The least significant difference (LSD) post hoc test was performed to identify which two groups showed significant differences. The statistical significance levels were set at *p* < 0.05. GraphPad prism software (8.0.2) (GraphPad Software, San Diego, CA, USA) was used for mapping.

## 3. Results

### 3.1. Jasmine Tea Intake Ameliorates the Behavioral Characteristics of Depressive-like Rats Induced by CUMS

According to Figure 2A,B, on day 0, there were no statistically significant differences in body weight among the three groups. The increase in body weight in rats was significantly decreased after 8 weeks of CUMS intervention when compared with the control group, while the body weight of CUMS-treated rats was increased after 4 weeks of jasmine tea administration. Meanwhile, depressive-like rats induced by CUMS were characterized by a decrease in food consumption and sugar preference during the 8-week external stimulation when compared with the control group, which could be improved after administration with jasmine tea (Figure 2C,D). As shown in Figure 2E, the immobility time of rats in CUMS group was significantly higher than that in the control group, which indicated that rats had no escape-oriented behavior and consciousness in the FST after CUMS intervention. However, the immobility time of CUMS rats declined after the jasmine tea administration. As shown in Figure 2F, the CUMS-treated rats showed a significant reduction in the number of inners, outers, and standing in OFT when compared with the control group, which indicated that those rats induced by CUMS expressed obvious depressive-like behavior. The number of inners, outers, and standing in OFT of CUMS-treated rats significantly increased after jasmine tea administration. Those results suggested that the depressive-like behaviors were significantly increased after 8 weeks of CUMS treatment, yet the administration with jasmine tea could alleviate depressive-like behaviors of CUMS-treated rats.

### 3.2. Jasmine Tea Intake Alleviates Neurotransmitters and Inflammatory Factors of Depressive-like Rats Induced by CUMS

The influences of jasmine tea on neurotransmitters (5-HT and BDNF) and pro-inflammations (TNF-α and NF-κB) in the hippocampus, cerebral cortex, colon, and serum of CUMS-treated rats were studied. The results showed that the contents of 5-HT and BDNF in the hippocampus, cerebral cortex, colon, and serum of the CUMS exposed rats were significantly declined when compared with the control group (Figure 3A,B). Jasmine tea administration significantly improved the contents of 5-HT and BDNF among those four tissues when compared with the model group (Figure 3A,B). The concentrations of TNF-α and NF-κB in the hippocampus, cerebral cortex, colon, and serum of Mod were much higher than those Con (Figure 3C,D), while jasmine tea inhibited the CUMS-induced elevation of the concentrations of TNF-α and NF-κB among different tissues (Figure 3C,D). These statistics indicated that jasmine tea administration could reverse the concentrations of neurotransmitters and pro-inflammatory cytokines in the CUMS-treated rats.

### 3.3. Jasmine Tea Intake Alleviates the Phenotype of the Hippocampus and Colon of Depressive-like Rats Induced by CUMS

HE and Nissl staining of the hippocampus is shown in Figure 4A,B). We found that hippocampal cells were arranged neatly, with a large number, high density, and complete nucleus in Con (Figure 4A,B), while the hippocampal cells were arranged unevenly and had nuclear pyknosis and degeneration, showing a loose shape, low density, reduced number, and larger cell gap after CUMS intervention (Figure 4A,B). Furthermore, the pyramidal cell layer in the model group becomes thinner. The hippocampus of those depression rats showed normal histopathological architecture, which was similar to the Con after the jasmine tea intervention (Figure 4A,B). According to the structure of the Con, we found that the colon mucosa was intact, epithelial cells were neatly arranged, and inflammatory cells rarely infiltrated (Figure 4C). The colon mucosa was absent, the lamina propria glands were damaged, goblet cells were reduced, and several inflammatory cells had infiltrated (Figure 4C). Those damages had obviously changed after the jasmine tea administration (Figure 4C).

### 3.4. The Amelioration of Gut Microbiota of Depressive-like Rats Induced by CUMS after Jasmine Tea Administration

The fecal microbial composition was analyzed using the 16S rRNA gene-sequencing method. In the discovery set, a total of 1,841,116 high-quality reads were obtained from all samples, with an average length of 102,284, which were aggregated into 2020 distinct operational taxonomic units (OTUs) at 97% sequence similarity. The analysis of the gut microbiota community revealed that findings on diversity measurements (Shannon and Simpson indices), as well as richness assessments (Chao1 and observed indices) in Mod, were significantly lower than those in the Con. At the same time, they could be improved after jasmine tea intervention (Figure 5A), which indicated that microbial richness and diversity indices in the Mod declined compared with Con, and this phenomenon of jasmine tea can alleviate microbial richness and diversity indices induced by CUMS. The results of the principal component analysis confirmed that the Mod had significantly different microbiota characteristics compared with the Con, and jasmine tea could regulate the gut microbiota in depression (Figure 5B). There were 82 genera among the three groups, while there was only 1 genus shared in Con and Mod and 7 genera shared in Con and NF, which also showed that the jasmine tea treatment significantly restored the intestinal microbes of CUMS-treated rats to the control one (Figure 5C). The relative abundance of gut microbiota at the phylum level was as follows (Figure 5E): the levels of *Firmicute* and *Bacteroidetes* in Con were 77.40% and 15.60%, respectively, and the ratio of F/B was 496.15%. The levels of *Firmicute* and *Bacteroidetes* in Mod were 80.71% and 12.27%, and the ratio of F/B was 658.78%. The levels of *Firmicute* and *Bacteroidetes* in NF were 80.02% and 13.98%, respectively, and the ratio of F/B was 572.39%, which indicated that jasmine tea could significantly repair the microbiota of CUMS-treated rats. The relative abundances of gut flora at the genus level were as follows (Figure 5D,F). CUMS significantly increased in *Ruminococcaceae, Erysipelotrichaceae, Verrucomicrobiaceae*, while decreased in *Clostridiaceae, Bacteroidaceae,* and *Peptostreptococcaceae*, and jasmine tea intake ameliorated those changes (Figure 5F). According to the circos circle of intestinal microbial based on the genus level (Figure 5D), compared with the Con (14.6%), the relative abundance of *Ruminococcus* was increased after CUMS induction (35.2%), which was 18.6% higher than that of Con. However, the relative abundance of *Ruminococcus* in CUMS-treated rats decreased to about 18.6% after the intervention of jasmine tea, which decreased by 14.6% compared with Mod. After CUMS treatment, the relative abundance of *Unspecified_Ruminococcaceae* (13.6%) was significantly reduced. However, the relative abundance in CUMS-treated rats of *Unspecified_Ruminococcaceae* increased to 16.4% after the intervention of jasmine tea, which was close to that in the control group.

A cladogram produced through LEfSe analysis of the gut microbiome dataset (Figure 6A,B) indicated that, at the genus level, there were 7 differentially abundant clades in Mod and 14 differentially abundant clades in NF (*p* < 0.05, LDA > 2.0). In total, 35 different genera were identified. There were 14 genera in Con, mainly included in *g__Blautia, o__Lactobacillales, f__Lactobacillaceae, g__Dorea, g__Lactobacillus, g__Coprobacillus, g__Klebsiella, g__Roseburia, f__Enterobacteriaceae, o__Enterobacteriales, o__Gemellales, c__Gammaproteobacteria, f__Gemellaceae,* and *g__Gemella*. The CUMS-treated rats were characterized by seven genera, including *g__Butyrivibrio, g__Bilophila, g__Anaeroplasma, o__Anaeroplasmatales, f__Anaeroplasmataceae, p__Tenericutes,* and *c__Mollicutes*. There were 14 genera in CUMS-treated rats after administration with jasmine tea, including *f__Clostridiaceae, f__Erysipelotrichaceae, c__Erysipelotrichi, o__Erysipelotrichales, g__Clostridium, g__Phascolarctobacterium, f__Veillonellaceae, g__Clostridium, f__Porphyromonadaceae, g__Parabacteroides, o__Pasteurellales, f__Pasteurellaceae, g__Aggregatibacter,* and *g__Holdemania*. The microbial composition was further determined by taxonomic profiling (Figure 7A,B). At the genus level (Figure 7C), the abundance of *Ruminococcus* and *Butyrivibrio* was significantly increased in Mod when compared with Con, but this phenomenon could be alleviated after the jasmine tea intervention. Moreover, the relative abundance of *Bacteroides, Blautia, Unspecified_Clostruduaceae, Clostridium, Lactobacillus*, and *Coprobacillus* in CUMS-treated rats was significantly increased after jasmine tea administration (Figure 7C).

### 3.5. Correlations of Neurotransmitters and Inflammation with Altered Gut Microbes in Depressive-like Rats Induced by CUMS Treated with Jasmine Tea

To illustrate the connection between the gut and the brain, a Spearman correlation analysis was conducted between intestinal microbes and phenotypic indicators of depression, including body weight, 5-HT, BDNF, TNF-α, NF-κB, etc. As shown in Figure 8A, pro-inflammatory factors such as TNF-α and NF-κB were distributed separately in the fourth quadrant and were strongly correlated with the model group, while factors such as body weight and neurotransmitter (5-HT, BDNF) were concentrated in the third and fourth quadrants and were highly correlated with normal and jasmine tea intervention groups. TNF-α and NF-κB were highly correlated with *Ruminococcus_1* and *Butyrivibrio*, while body weight and neurotransmitters (5-HT, BDNF) were highly correlated with *Blautia, Unspecified_Clostrudiales, Unspecified_Ruminococcaceae* and other microorganisms (Figure 8B).

The results of the Spearman correlation showed that *Butyrivibrio, Anaeroplasma Unspecified_Veillonellaceae*, etc., were significantly negatively correlated with body weight 5-HT and BDNF and significantly positively correlated with inflammatory factors TNF-α and NF-κB, while *Unspecified_Christensenellaceae, Clostridium_2,* and *Unspecified_Coriobacteriaceae* showed a strongly positive relationship with body weight 5-HT and BDNF and showed a strongly negative relationship with TNF-α and NF-κB.

### 3.6. The Alleviation of Blood Metabolisms in Depressive-like Rats Induced by CUMS after Jasmine Tea Administration

Since the gut microbiota plays a consistent role in controlling the metabolic processes of the host, the serum metabolome is seen as a practical reflection of the gut microbiome. To evaluate the influence of jasmine tea intervention on metabolite formation, the metabolites in rat serum were analyzed by UHPLC-MS/MS. The Partial Least Squares Discriminant Analysis (PLS-DA) method was used to analyze the separation among the three groups, and it was found that the metabolites associated with CUMS-treated rats and CUMS-treated rats companied with jasmine tea were changed (Figure 9A–C). In order to better differentiate these particular distinctions, we employed a supervised multivariate OPLS-DA model to elucidate the various variables. According to the analysis results, the model group and the control group were separated (Figure 9D). It is worth noting that NF is also separated from Mod (Figure 9E), indicating that jasmine tea administration improved metabolic disorders of CUMS-treated rats. Furthermore, 628 metabolites were identified through online databases. A total of 100 distinct metabolites were found when comparing Con and Mod, of which 44 were upward and 56 were downward after CUMS induction (Appendix A, VIP > 1.0 and *p*-values < 0.05). More importantly, there were 30 different metabolites identified during the comparison between Mod and NF, of which 22 were upward and 18 were downward after jasmine tea intervention (Appendix A, VIP > 1.0 and *p*-values < 0.05). The amounts of theobromine, SM (d17:0/24:1), PC (20:2/20:3), Cer-NS (d18:1/24:0), Cer-NS (d18:1/22:0), theophylline, and 17α-Ethynylestradiol were increased after jasmine tea treatment (Appendix A). Meanwhile, PC (18:2e/2:0), PC (21:2/20:5), Ecgonine methyl ester, PC (18:3e/22:1), γ-Linolenic acid ethyl ester, and PC (14:1e/20:1) were decreased after jasmine tea treatment (Appendix A). The key metabolites (VIP > 2) were mainly included in theobromine, SM (d17:0/24:1), Tetrahydrocortisone, PC (20:2/20:3), etc. (Appendix A). The jasmine tea intervention regulated the metabolites to Con, including DL-Tryptophan, Inole, Lysopc 18:2, PC (19:2/19:2), PC (20:3/19:2), PC (16:2e/2:0), LPC22:6, Cer-NS (d18:1/24:0), etc. (Appendix A), which were annotated through the KEGG database and were successfully annotated to steroids, peptides, lipids, nucleic acids, vitamins and cofactors, hormones and transmitters, and antibiotics (Figure 9F). According to the mean decrease in accuracy, where they contributed to the differentiation in samples, the random forest analysis chart showed that their contribution to the sample differentiation was greater (Appendix A). Three metabolic pathways with impact values > 0 and *p* < 0.05 showed enrichment between the Mod and NF conditions (Appendix A), including glycerophospholipid metabolism pathways, glycine serine and threonine metabolism pathways, and nicotinate and nicotinamide metabolism pathways (Figure 9G).

### 3.7. Correlations Among Neurotransmitters, Inflammation, and Serum Metabolites in Depressive-like Rats Induced by CUMS Treated with Jasmine Tea

To further study the potential correlations among neurotransmitters, inflammation, and the serum metabolome, a Spearman correlation analysis was conducted (Figure 10). The analysis revealed that the neurotransmitters, inflammation, and metabolites in the serum samples had a correlation. The results showed that PC (16:0/16:1), 3,4-hydroxy-3- methoxyphenyl-propanoic acid, N-4-fluorophenyl-N-2-piperidinophenyl urea, etc., were significantly negatively correlated with body weight, 5-HT, and BDNF, while those metabolites were highly positively correlated with TNF-α and NF-κB. Moreover, PC (18:5e/22:6), tetrahydrocortisone, PC (14:0e/3:0), 2-oxindole, etc., showed a strongly positive relationship with body weight, 5-HT, and BDNF, and showed a strongly negative relationship with TNF-α and NF-κB.

### 3.8. Correlations between Gut Microbes and Serum Metabolites in Depressive-like Behavior Induced by CUMS

To further study the potential correlations between intestinal microbiome and serum metabolome, a correlation analysis was conducted (Figure 11 and Appendix A). The analysis indicated that there was a correlation between the gut microbiota and metabolites found in the serum samples. It was found that 15 genera were significantly associated with a range of differential metabolites, including *Lachnospiraceae_Clostridium, Phascolarctobacterium, Anaeroplasma, Klebsiella, Bilophila, Butyrivibrio, Gemella, Roseburia, Coprobacillus, Dorea, Lactobacillus, Clostridium, Parabacteroides, Holdemania, and Blautia*. Among those metabolites, theophylline and theobromine were negatively correlated with *Lachnospiraceae_Clostridium* and *Phascolarctobacterium*, while they positively correlated with *Bilophila*.

## 4. Discussion

The influence of jasmine tea in preventing depression has been demonstrated by experiments. However, whether and how jasmine tea alleviated dysbiosis and changes in metabolites caused by CUMS remained unclear. The objective of this study is to investigate the potential mechanism of jasmine tea in preventing depression from the perspective of intestinal flora and metabolites.

The main features of depression mainly include an unresponsiveness to situations; a persistent inability to feel pleasure; a unique despondent mood; and common physical signs like changes in appetite or weight, as well as early morning awakening and poor morning mood [34]. Anhedonia is a central symptom of depression, mainly manifested in SPT and food consumption [1]. This study evaluated the preventive effect of jasmine tea on depression through a variety of traditional behavioral tests. Depressive-like rats induced by CUMS were characterized by decreasing body weight, food consumption, and sugar preference during 8 weeks when compared with the control group. The number of inners, outers, and standing in the model group significantly decreased compared to the control group. The duration of immobility in the FST was significantly longer in the model group compared to the control group. Those depressive behavioral performances were significantly restored after the jasmine tea administration.

There is a strong link between the emotion-regulating regions of the brain (such as the prefrontal cortex, hippocampus, limbic system, etc.) and serotonin in the center of the nervous system, which constitutes the physiological basis for the 5-HT nervous system to participate in regulating anxiety and (or) depression [35]. The BDNF is an important factor in forming new connections between neurons and the brain factor trophic origin [36]. The over-expression of FXR in the hippocampus in naïve rats led to depression-like symptoms and reduced the expression of BDNF in the hippocampus [37]. The BDNF level in the plasma of untreated patients was also much lower than in the treated and control group [37]. In this study, the results showed that the concentrations of 5-HT and BDNF in the hippocampus, cerebral cortex, colon, and serum of the CUMS-exposed rats were significantly declined when compared to the control group. Jasmine tea administration significantly increased the contents of 5-HT and BDNF in all four tissues when compared with the model group. Increasing the pro-inflammatory cytokines, such as TNF-α and NF-κB, is a crucial manifestation of depression [38,39]. The pro-inflammatory mediators, including leukotrienes, prostaglandins, and cytokines, showed a higher level in depressed people when compared with healthy ones. We found that TNF-α and NF-κB expression in the hippocampus, cerebral cortex, colon, and serum were significantly elevated in the model group compared with the control one, while jasmine tea inhibited the CUMS-induced elevation of TNF-α and NF-κB expression in the hippocampus, cerebral cortex, colon, and serum.

Hippocampus and colon homeostasis is maintained by the crosstalk among the neurotransmitters, immune system, and gut microbiota. An imbalance in neurotransmitters, immune system, and gut flora often causes the destruction of the construction in the hippocampus and colon. He et al. [40] found that gut microbiota can induce microglial priming in the dentate gyrus after stress induction, which is related to a hyper-immune response to stress and impaired hippocampal neurogenesis. In this study, the HE and nissl staining of the hippocampus indicated that CUMS-exposed rats had nuclear pyknosis and degeneration and had a higher level of inflammation compared with the Con. The colon mucosa was obviously absent, the lamina propria glands were damaged, the number of goblet cells was decreased, and multiple inflammatory cells infiltrated after CUMS induction. Those damages had obviously been restored after the jasmine tea administration.

Particularly, dysbiosis is regarded as the pathogenesis of depression, and the microbiome is considered a crucial factor mediating depression [12,41]. Furthermore, we utilized 16S rRNA gene-sequencing technology to assess the composition, richness, and diversity of the fecal microbiota. Our results indicated a notable distinction in the microbial profile between CUMS-induced depressive rats and the Con, which was closely correlated with their behavioral patterns. The abundant diversity and richness in the gut microbiota are recognized as vital indicators of host health [42]. Studies have indicated that CUMS intervention altered the composition and diversity of gut microbiota and enhanced the permeability of the intestinal epithelium, indicating a defect in the intestinal barrier [43,44]. We found that CUMS caused a lower microbial composition richness and diversity indices than Con, which suggested that the physiological processes of depressive-like rats were disordered, while the richness and diversity indices of depressive-like rats were increased to improve the physiological processes after jasmine tea treatment. Mice that ingested large amounts of sucrose significantly decreased the amounts of *Bacteroidetes* and enhanced the amounts of *Firmicutes*, *Proteobacteria,* and pathogenic *Helicobacteraceae* [45]. Rtl et al. found that MDD led to reduced levels of *Firmicutes* and elevated levels of *Bacteroidetes*, showing similar patterns at the class level (*Clostridia* and *Bacteroides*) and order level (*Clostridiales* and *Bacteroidales*) [46]. In contrast, studies found that rats induced by CUMS for 3 weeks significantly changed in intrathecal beta-diversity, characterized by a crucial improvement in the *Firmicutes/Bacteroidetes* (F/B) ratio due to a reduced *Bacteroidetes* and an enhanced *Firmicutes* relative abundance [47,48]. These changes were associated with reductions in the Porphyromonadaceae family, specifically in the genus *Barnesiella*, which is a crucial member of gut microbiota in mice and humans and has been described as having protective properties [47]. In this study, the results showed that the ratio of F/B was significantly enhanced after CUMS induction, while jasmine tea ameliorated the relative abundance of *Firmicute* and *Bacteroidetes*, leading to a lower F/B ratio, which illustrated that jasmine tea administration could significantly restore the microbiota in the depressive rats. *Lactobacillus* could maintain a stable immunity, reduce fecal enzyme activity, and reduce viral diarrhea [49,50]. Tianyi Liquid could enhance the relative levels of *Lactococcus, Ruminococcaceae, Lachnospiraceae_NK4A136_group, and Lactobacillus* in depressive rats [51]. We found that the levels of *Ruminococcus* and *Butyrivibrio* were significantly elevated in Mod compared with Con, but this phenomenon could be alleviated after the jasmine tea intervention. Moreover, jasmine tea administration significantly improved the relative abundance of *Bacteroides, Blautia, Unspecified_Clostruduaceae, Clostridium, Lactobacillus, and Coprobacillus* in depressive rats induced by CUMS. The intestinal microflora plays a crucial role in regulating brain functions and behavior. The gut microbiota is essential in adjusting cerebral functions and behavior. The relative abundance of *Odoribacter, Ruminococcus, Lactobacillus, Mucispirillum, Helicobacter, and Oscillospira* generally seemed to be specially related to depressive-like expression and depressive-related indexes [52]. The expression of DA and 5-HT was changed by *Lactobacillus* in the intestinal tract via the nerve [53]. The gut microorganisms also affect the immune and nervous systems. The gut microbiota can impact the production of pro-inflammatory cytokines and the function of the nervous system by playing a role in neurotransmitter synthesis [10]. In this study, we found that *Butyrivibrio, Anaeroplasma Unspecified_Veillonellaceae*, etc., were significantly negatively correlated with body weight 5-HT and BDNF while significantly positively correlated with inflammatory factors TNF-α and NF-κB, while *Unspecified_Christensenellaceae, Clostridium_2,* and *Unspecified_Coriobacteriaceae* had significantly positive relationships with body weight 5-HT and BDNF and had negative relationships with TNF-α and NF-κB.

The microbiome regulates the central nervous system (CNS) activity via serum metabolites [54]. Serum metabolic profiles were analyzed using UHPLC-MS/MS metabolomics in this study. Multivariate data analysis was used to identify the differential metabolites and analyze the changes in serum metabolites. The crucial metabolites in depression mainly include lipopolysaccharide (LPS), Lysophospholipids (LysoPCs), short-chain fatty acids (SCFAs) and phospholipids (PCs, PEs, and PSs), and sphingomyelin (SMs). Phenylalanine, uric acid, cholic acid, tryptophan, and lysophosphatidyl choline have been identified as possible biomarkers related to the mechanism of depression [55]. The results declared that jasmine tea treatment could reverse the disturbances of serum metabolomics in CUMS’s rats. Theobromine and theophylline played a vital role in easing depressive symptoms. We found that the amounts of theobromine, SM (d17:0/24:1), PC (20:2/20:3), Cer-NS (d18:1/24:0), Cer-NS (d18:1/22:0), theophylline, and 17α-Ethynylestradiol were increased after jasmine tea treatment. Meanwhile, PC (18:2e/2:0), PC (21:2/20:5), Ecgonine methyl ester, PC (18:3e/22:1), γ-Linolenic acid ethyl ester, and PC (14:1e/20:1) were decreased after jasmine tea treatment.

Melancholic depression is a chronic stress state that activates cortisol, corticotropin-releasing norepinephrine (NE), and hormone (CRH) pathways without inhibitory feedback mechanisms. Chronic cortisol levels mediated immunosuppression in depression and inflammatory activation in atypical depression. The analysis of the metabolites in serum, brain, and urine revealed that SJ and CPM primarily improved alterations in monoamine neurotransmitter metabolites, while GS affected both excitatory/inhibitory monoamine neurotransmitters and amino acid metabolites. Thirty-three metabolites of small molecular markers in the whole serum of dementia patients were indicated when compared with healthy elderly subjects. Kynurenine, quinolinic acid, and indoxyl-sulfate, which may be a neurotoxin in the central nervous system, had increased.

The key metabolic pathways related to the amelioration of depressive-like symptoms by jasmine tea were further revealed by enrichment analysis. Importantly, the changes in gut microbiotas were highly related to a range of metabolites. Our findings illustrated that the intestinal bacteria might have a significant impact on how jasmine tea can help relieve symptoms of depression, of which the mechanism may have a closer relationship with the regulation of riboflavin metabolism, caffeine metabolism, nicotinate, and nicotinamide metabolism.

The metabolites were found to have an inverse relationship with the host’s physiology [56]. The sphingomyelins were also found to have a closer relationship with depression symptoms [50]. We found that PC (16:0/16:1), 3,4-hydroxy-3-methoxyphenyl-propanoic acid, N-4-fluorophenyl-N-2-piperidinophenyl urea, etc., were significantly negatively correlated with body weight, 5-HT, and BDNF and had a positive relationship with inflammatory factors TNF-α and NF-κB. In contrast, PC (18:5e/22:6), tetrahydrocortisone, PC (14:0e/3:0), 2-oxindole, etc., had a significant positive relationship with body weight, 5-HT, and BDNF and had a negative relationship with TNF-α and NF-κB. The contents of quinoline, indoleamine 2,3-dioxygenase, tryptophan, and the KYN/TRP ratio were reduced, and the levels of kynurenic acid and 5-HT were improved after being treated with Tiansi Liquid [51]. It was indicated that the relative abundance of the *Lachnospiraceae_NK4A136_group* was negatively correlated with the content of quinoline. Moreover, the relationship between the key microbiota and differential metabolites was examined in this study. We can see that there were 15 genera, including *Lachnospiraceae_Clostridium, Phascolarctobacterium, Anaeroplasma, Klebsiella, Bilophila, Butyrivibrio, Gemella, Roseburia, Coprobacillus, Dorea, Lactobacillus, Clostridium, Parabacteroides, Holdemania*, and *Blautia*, that were significantly correlated with a range of differential metabolites. Among those metabolites, theophylline and theobromine were negatively correlated with *Lachnospiraceae_Clostridium* and *Phascolarctobacterium* and positively correlated with *Bilophila*. Above all, the results indicated that the gut microbiota and serum metabolome of CUMS-treated rats were disturbed, and jasmine tea intervention could repair the gut microbiome and serum metabolome of CUMS-treated rats.

However, there were also some limitations: (1) While the evidence that jasmine tea can restore gut microbiota imbalance was provided, the causality can be confirmed by fecal transplantation experiments. (2) This research only used male rats as material. Women have a high possibility of falling into depression due to the interference of external factors. We can use female rats as material to carry out future experiments. (3) The shotgun metagenomic sequencing approach is capable of differentiating the distinct bacterial strains present in depressive-like rats treated with jasmine tea, given the constraints associated with the 16S rRNA sequencing method. (4) In regard to the metabolic pathways associated with improving depression through jasmine tea, it is essential to further investigate their regulatory targets. (5) Due to the diversity of the microbiome, the results obtained from the rat model need to be carefully applied directly to humans. Therefore, more exploration is still needed, especially in human samples.

## 5. Conclusions

Jasmine tea attenuated weight loss and depressive-like behavior in CUMS-treated rats. Such a prevention-depressant effect was partially attributed to jasmine tea increasing neurotransmitters (5-HT and BDNF), suppressing pro-inflammatory cytokines (TNF-α and NF-κB), and restructuring the gut microbiome and serum metabolism. We used multi-omics data to outline the bacteria and serum metabolites landscape of depressive-like rats induced by CUMS after being treated with jasmine tea. We found that intestinal flora may participate in the prevention of depression by jasmine tea, including *Bacteroides, Blautia, Clostridium, Lactobacillus, Ruminococcus, Butyrivibrio*, and the metabolites disturbance caused by CUMS may be regulated through riboflavin metabolism, caffeine metabolism, and nicotinate and nicotinamide metabolism. These results can provide a new perspective that jasmine tea has a positive influence on depression treatment for understanding the mechanism of ameliorating depression induced by CUMS after administrated jasmine tea, which can provide a potential strategy for the prevention of depression through jasmine tea administration.

## Figures and Tables

**Figure 1 foods-13-02636-f001:**
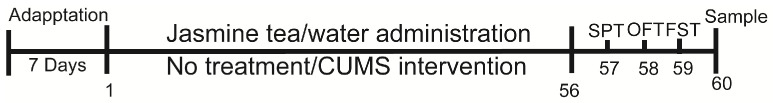
The schedule of experimental procedures. SPT, sugar preference test; OFT, open-field test; SFT, forced swimming test.

**Figure 2 foods-13-02636-f002:**
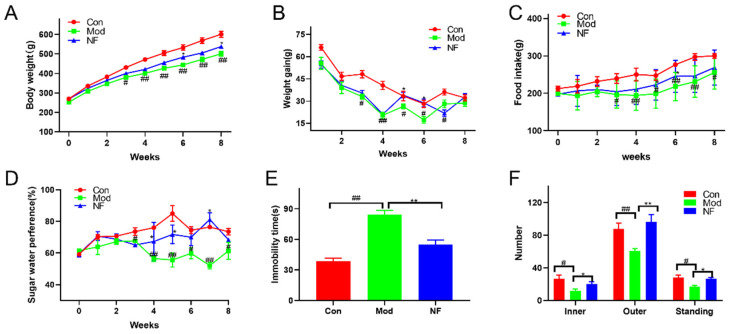
Results of the behavioral tests. (**A**) Body weight during the experiment. (**B**) Body weight gain during the experiment. (**C**) Food intake during the experiment. (**D**) Sugar preference during the experiment. (**E**) Immobility time comparison of FST. (**F**) The comparison of number of inners, outers, and standing in OFT. Con, control; Mod, model; NF, CUMS treated with jasmine tea. SPT, sugar preference test; FST, force swimming test; OFT, open field test. # *p* < 0.05, ## *p* < 0.01 versus the control group; * *p* < 0.05, ** *p* < 0.01 versus the model group.

**Figure 3 foods-13-02636-f003:**
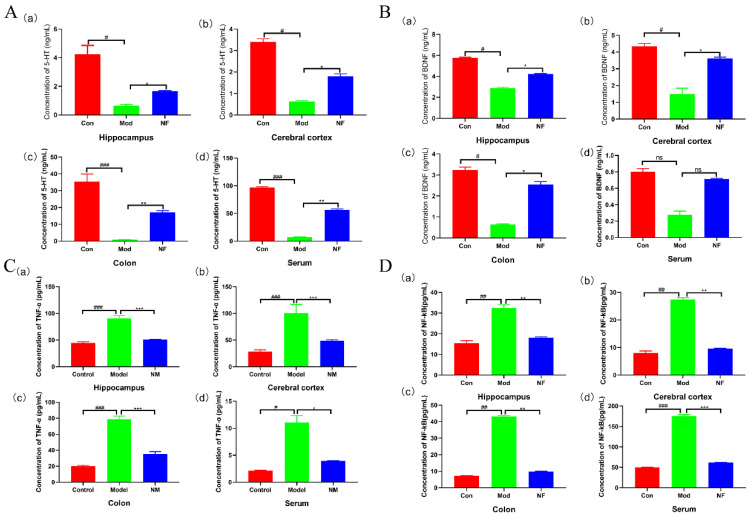
The levels of neurotransmitter and inflammation factors among different tissues. (**A**) 5-HT; (**B**) BDNF; (**C**) TNF-α; (**D**) NF-κB. # *p* < 0.05, ## *p* < 0.01, ### *p* < 0.01 versus the control group; * *p* < 0.05, ** *p* < 0.01, *** *p* < 0.001 versus the model group.

**Figure 4 foods-13-02636-f004:**
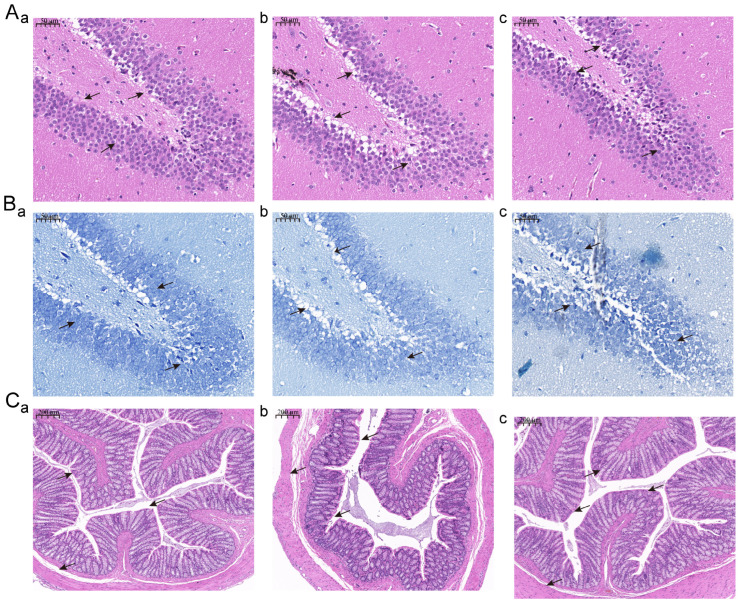
The ameliorations of phenotype in hippocampus and colon of CUMS-induced rats after jasmine tea administration. (**A**) HE staining of the hippocampus. (**B**) Nissl staining of the hippocampus. (**C**) HE staining of the colon. →, the comparison of special structures in the figure.

**Figure 5 foods-13-02636-f005:**
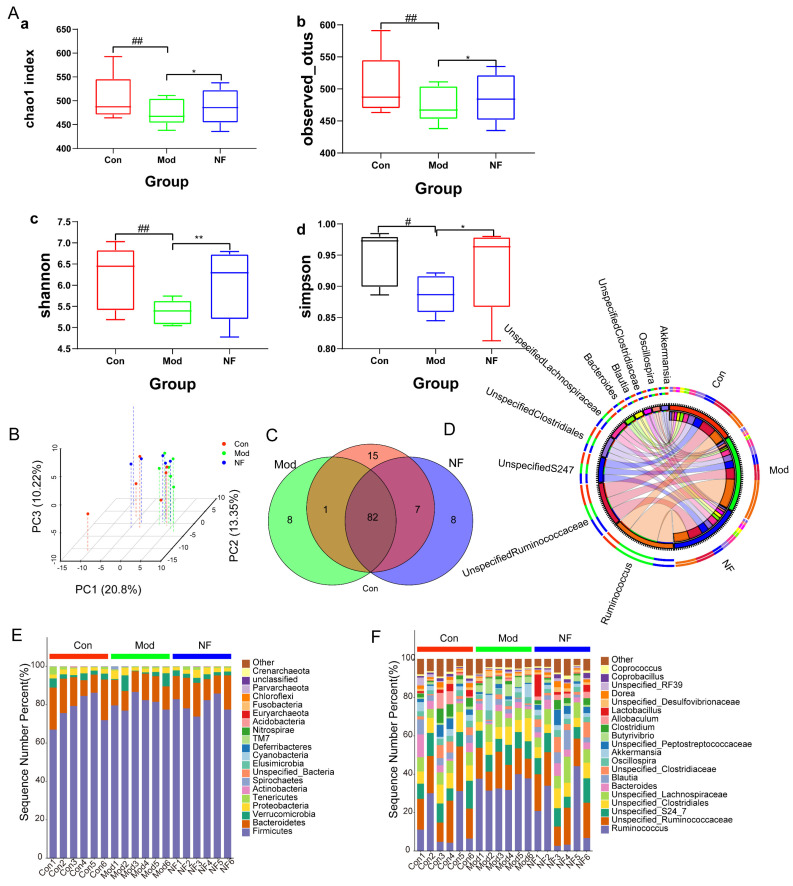
Gut microbial characteristics of Con, Mod, and NF. (**A**) Alpha-phylogenetic diversity analysis showed that depressive-like rats induced by CUMS had lower microbial richness in four indexes relative to controls; these indexes had increased when administrated with jasmine tea. (**B**) Principal component analysis (PCA) revealed that the gut microbiome composition in rats with depressive-like symptoms induced by CUMS was markedly distinct from that of Con, and the intestinal microbial composition of depressive-like induced by CUMS was restored after being treated with jasmine tea. (**C**) Venn diagram for taxonomy of gut microbes based on genus level. (**D**) Circus circle diagram of the top 10 relative abundances in intestinal microbial classification based on genus level. (**E**) The gut microbiota compositions among the experimental groups at the phylum level. (**F**) The relative abundance of gut microbes at the genus level. # *p* < 0.05, ## *p* < 0.01 versus the control group; * *p* < 0.05, ** *p* < 0.01, versus the model group.

**Figure 6 foods-13-02636-f006:**
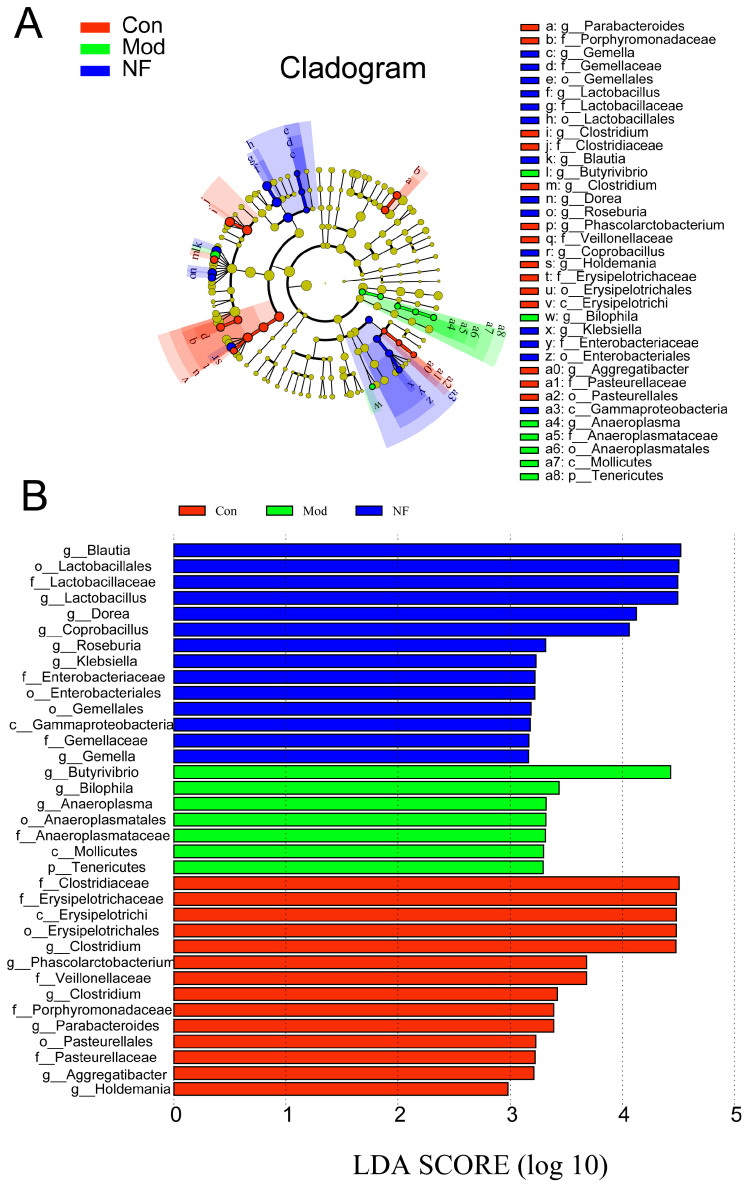
Linear discriminant analysis effect size (LEfSe) analysis was conducted with a threshold of LDA > 2.0. The results were visualized using a cladogram (**A**) and a histogram (**B**) and showed 35 genera responsible for discriminating in Con, Mod, and NF.

**Figure 7 foods-13-02636-f007:**
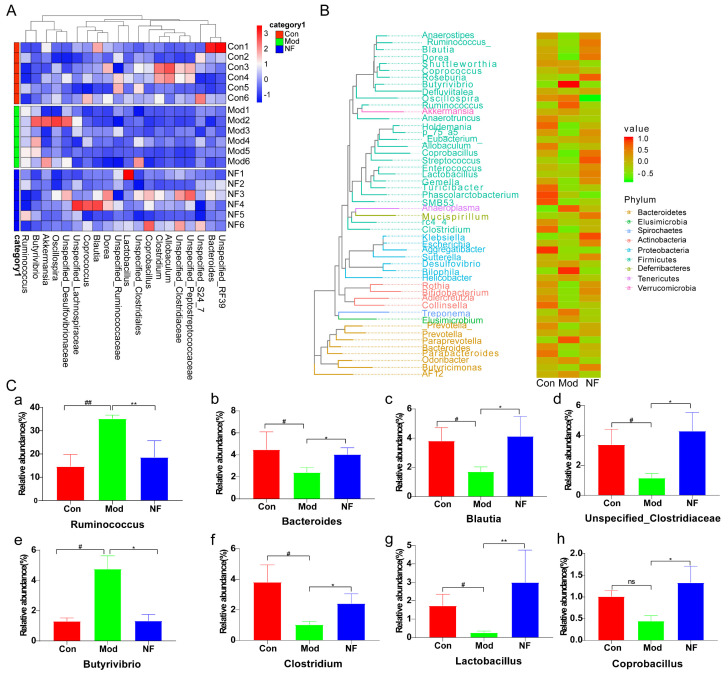
Jasmine tea intervention regulated the gut microbiota in CUMS-induced depressive rats. (**A**) Heatmap of the gut microbiome in genus levels. (**B**) Phylogenetic tree and heatmap of abundances distribution between groups in phylum level. (**C**) The comparison of relative abundance of gut microbial community members was conducted at the genus level among three groups. Data represent the mean ± SEM of six rats in each group. # *p* < 0.05, ## *p* < 0.01 versus the control group; * *p* < 0.05, ** *p* < 0.01 versus the model group.

**Figure 8 foods-13-02636-f008:**
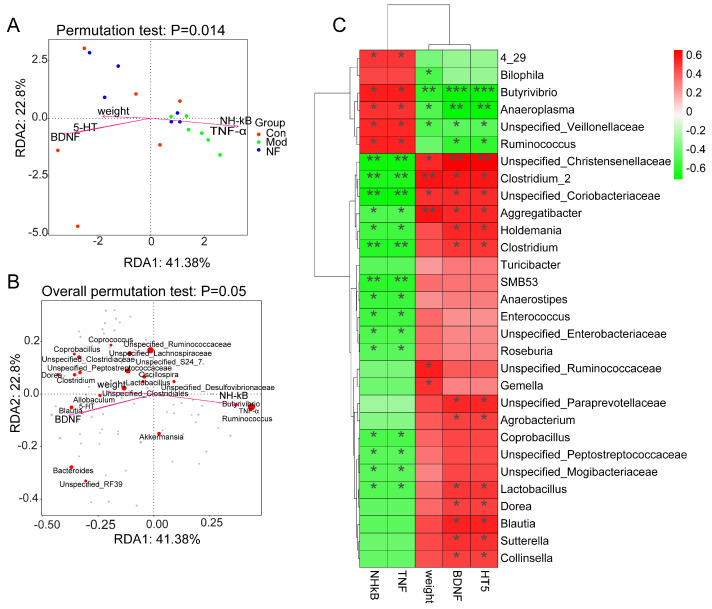
Associations of gut microbes with neurotransmitters and inflammation factors based on Spearman correlation analysis. (**A**) The composition of intestinal microorganisms based on the genus level and environmental factors being analyzed through a mapping test. (**B**) Overreplacement test map of intestinal microorganisms based on generic levels and environmental factors. (**C**) Visualization of Spearman’s rank correlation in the form of a heatmap of 30 genera and neurotransmitters, with inflammations in colon. * *p* < 0.05, ** *p* < 0.01, *** *p* < 0.001.

**Figure 9 foods-13-02636-f009:**
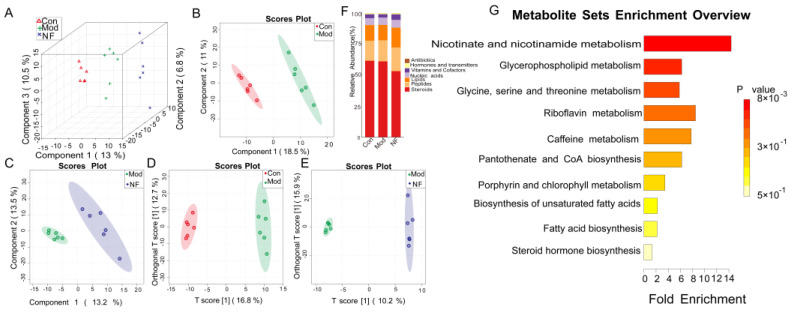
Multivariate analysis of metabolic profiles of Con, Mod, and NF. (**A**–**C**) The evaluation scatter plots of different groups from the PLS-DA data. (**D**) Con and Mod pairwise comparison OPLS-DA evaluation scatter plots. (**E**) Mod and NF pairwise comparison OPLS-DA evaluation scatter plots. (**G**) Stacked column chart of the percentage of metabolites that play a biological role. (**F**) Pathway topology enrichment among different treatments.

**Figure 10 foods-13-02636-f010:**
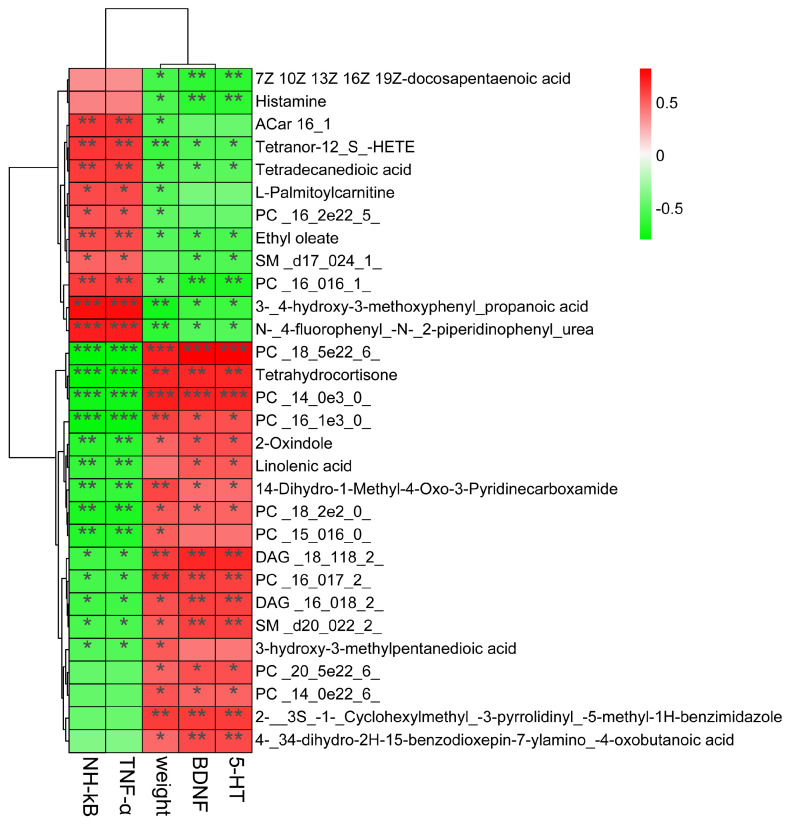
Associations of gut microbes with neurotransmitters and inflammation factors based on Spearman correlation analysis. * *p* < 0.05, ** *p* < 0.01, *** *p* < 0.001.

**Figure 11 foods-13-02636-f011:**
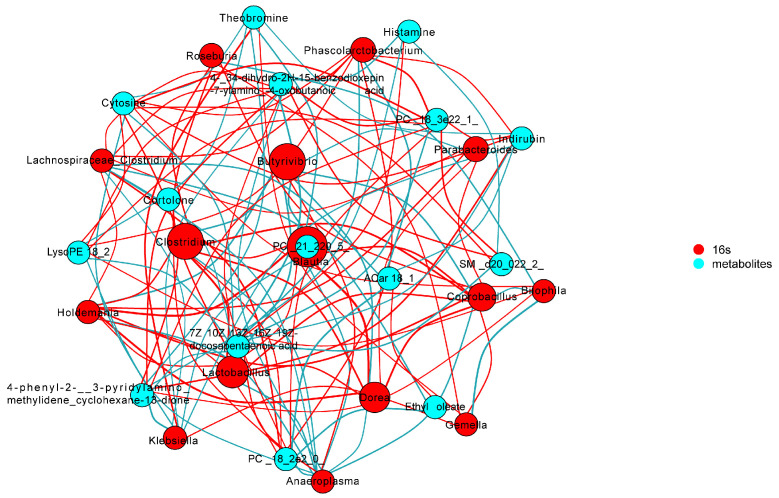
Network diagram of associations of gut based on lefse-genus and metabolites based on OPLS-DA.

## Data Availability

The original contributions presented in the study are included in the article/Appendix A, further inquiries can be directed to the corresponding author.

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
