# Peer review of "An Analysis of the Intestinal Microbiome Combined with Metabolomics to Explore the Mechanism of How Jasmine Tea Improves Depression in CUMS-Treated Rats"

_foods, 2024, doi:10.3390/foods13162636_

Round 1

Reviewer 1 Report

Comments and Suggestions for Authors

If the aim of the paper was to study "The intestinal microbiome combined with metabolomics to explore the mechanism of jasmine tea improving depression in CUMS rats, the authors should explain why they decided that jasmine tea was made with jasmine flower and green tea and also said why a single concentration was chosen? Why did they not use different concentrations of jasmine tea? Why did they not use experimental design?

On the other hand, the possible bioactive compounds of jasmine tea that attenuate depressive-like behavior are unclear.

The conclusion does not highlight the results obtained.

The references do not follow a format.

COMMENTS

Please review the document because there are specific corrections

Linea 38.- Check the translation in the sentence “of more and more people”

Línea 41.- The phrase “Disruption of the gut microbiota” could be placed for Dysbiosis.   

Línea 44.- “Studies indicated that the disturbances of the gut microbiome homeostasis” can be repaced by “Studies indicated that the dysbiosis”

Línea 45-47.- Could be add “denominated microbiota-gut-brain axis”.

Línea 91-92.-  “There is increasing evidence showing that gut microbiota is involved in the development of depression by modulating metabolism via the gut-brain axis” It is repetitive, it was mentioned in the previous paragraphs.

Línea 117.- What does  NF means?

Línea 121.- Why was an experimental design not used with variations in the dose of the tea?

Línea 137.- Please increase the image quality

Línea 181.- What sequencing method was used? It is not mentioned in the text.

Línea 203.- What Rstudio update was used, in addition to the “vegan” packaging that others were used?

Línea 232.- What normalization method was used for the metabolomic data?

Línea 223.- With which database were the data obtained from HPLC/Masses compared?

Línea 283. What does  Con, Mod y NF means?

Línea 307. There are many figures or graphs, this type of results can be summarized with another type of data analysis, for example, non-parametric analysis such as principal components or multivariate.

Línea 335- 360. They describe graphs A-D but do not mention whether or not there is a significant difference in the analyzes carried out.

Línea 433.- Please explain te meaning of  “*” in the heatmap, some * cannot be seen, improve color contrast.

Línea 444-447.- The authors said there are not changes in the presence of metabolites, please put the statistical analysis

Línea 549-558. Please discuss the results obtained with other authors. If what they got is correct or similar with some other type of food.

Línea 564-566.- The text said “The high level of diversity and richness of gut microbiota is considered as the healthy factor of the host”, but specifically which bacteria are related to the health of the host. Mention related bacteria.

Línea 656.- Please explain why you compare only with theophylline and theobromine.

Author Response

Response to Reviewer Comments

Dear Editors and Reviewers:

Thank you very much for your letter and for the reviewers’ comments concerning our manuscript entitled “The intestinal microbiome combined with metabolomics to explore the mechanism of jasmine tea improving depression in CUMS rats”. Those comments are all valuable and very helpful for revising and improving our paper. We have seriously taken the comments into consideration in revising our manuscript. Revised portion are marked in red in our manuscript. The main corrections in the paper and the responds to comments are as follows:

Reviewer1:

  1. Comments and Suggestions for Authors If the aim of the paper was to study "The intestinal microbiome combined with metabolomics to explore the mechanism of jasmine tea improving depression in CUMS rats, the authors should explain why they decided that jasmine tea was made with jasmine flower and green tea and also said why a single concentration was chosen? Why did they not use different concentrations of jasmine tea? Why did they not use experimental design?

Response: Thanks for the positive comments and valuable suggestions. Here, we will answer your question. Firstly, jasmine flower belongs to "temperament flower" and is the raw material for scenting scented tea and extracting essential oil. Jasmine tea is a unique type of reprocessed tea in China and has great market potential in China and Southeast Asian countries. In the traditional process, green tea is often used as tea dhool to scenting jasmine tea to obtain jasmine tea. In recent years, our team has carried out a new process optimization of jasmine tea scenting process in order to obtain better quality jasmine tea, which was published on food chemistry (Study on the key volatile compounds and aroma quality of jasmine tea with different scenting technology), molecules (Analysis of Volatile Components of Jasmine and Jasmine Tea during Scenting Process), Journal of Tea Sciences (Study on the characteristic aroma components of jasmine tea), etc. So, the jasmine tea produced by new process was used as the material to explore the influence on ameliorating depression induced by CUMS, which can provide some basis for the market promotion of jasmine tea.

In our previous study (Jasmine Tea Attenuates Chronic Unpredictable Mild Stress induced Depressive-Like Behavior in Rat Via the Gut-Brain Axis, which was published on nutrients), three concentrations of jasmine tea, including the high, medium and low, were screened, and we found that the medium dose of jasmine tea had a better effect on relieving depression caused by CUMS. Therefore, in this study, we only selected medium-dose of jasmine tea for related research.

2.On the other hand, the possible bioactive compounds of jasmine tea that attenuate depressive-like behavior are unclear.

Response: Jasmine tea is scented by jasmine flowers and green tea, which contains not only tea polyphenols, L-theanine, flavonoids, and other bioactive components of tea, but also the pleasant aroma from jasmine. Those bioactive compounds may have a positive effect on alleviating depression. In the next study, we will isolate the effective components from jasmine tea and further explore which components play a positive role in alleviating depression.

COMMENTS 

Please review the document because there are specific corrections 

Line 38.- Check the translation in the sentence “of more and more people”

Response: Thank you for your valuable suggestion. We are very sorry for our negligence of this mistake. The correct statement is “more and more people”. We had modified in the manuscript.

Line 41.- The phrase “Disruption of the gut microbiota” could be placed for Dysbiosis.

Response: Thank you for your valuable suggestion. We have made correction according to the reviewer’s comments. We had replaced the “Disruption of the gut microbiota” to “Dysbiosis”.

Line 44.- “Studies indicated that the disturbances of the gut microbiome homeostasis” can be repaced by “Studies indicated that the dysbiosis”

Response: Thank you for your valuable suggestion. We have made correction according to the reviewer’s comments. We had replaced the “Studies indicated that the disturbances of the gut microbiome homeostasis” to “Studies indicated that the dysbiosis”.

Line 45-47.- Could be add “denominated microbiota-gut-brain axis”.

Response: Thank you for your valuable suggestion. We had added the “denominated microbiota-gut-brain axis” in Line 49-50.

Line 91-92. -“There is increasing evidence showing that gut microbiota is involved in the development of depression by modulating metabolism via the gut-brain axis” It is repetitive, it was mentioned in the previous paragraphs.

Response: Thank you for your valuable suggestion. We had deleted this statement in Line 91-92.

Line 117.- What does NF means?

Response: NF means CUMS rats treated with jasmine tea scented by new process in middle dose.

Line 121.- Why was an experimental design not used with variations in the dose of the tea?

Response: In our previous study (Jasmine Tea Attenuates Chronic Unpredictable Mild Stress-induced Depressive-Like Behavior in Rat Via the Gut-Brain Axis, which was published on nutrients), three concentrations of jasmine tea, including the high, medium and low, were screened, and we found that the medium dose of jasmine tea had a better effect on relieving depression caused by CUMS. Therefore, in this study, we only selected medium-dose of jasmine tea for related research.

Line 137.- Please increase the image quality 

Response: Thanks for your valuable suggestion. We had improved the image quality.

Line 181.- What sequencing method was used? It is not mentioned in the text. 

Response: In this study, we had used 16S rRNA sequencing to analysis the gut microbiota in feces, which were mentioned in Line 183.

Line 203.- What Rstudio update was used, in addition to the “vegan” packaging that others were used?

Response: We had used the Rstudio packagings, including Phyloseq, mixOmics, ggtree, pheatmap, dunn.test, agricolae, etc. to analyze the 16S rRNA sequencing statistic.

Line 232.- What normalization method was used for the metabolomic data?

Response: The normalization method was included in three steps:

First, in-sample correction: that is, the abundance of all features in the sample divided by the median abundance of the sample (similar to relative abundance calculation); This is to correct library size. In the process of measurement, the total metabolite content of each sample varies greatly, which is usually the systematic error caused by sample collection and measurement. Dividing by the median, average, or sum is a common way to correct such systematic error.

Second, content matrix correction: log conversion of all content values; T-test and difference comparison methods such as ANOVA require the metabolite content to follow normal distribution, so we generally use log transformation to make the metabolite content distribution close to normal distribution

Third: intra-feature correction: that is, the abundance of all samples corresponding to the feature minus the mean abundance of the feature and then divided by the standard deviation of the abundance of the feature. The purpose of correction within feature is to make the mean and standard deviation (or mean, quartile, scale) of all metabolites at the same level; In PCA, PLSDA, OPLSDA and machine learning analyses, if the metabolites with high mean and standard deviation are not standardized, the importance of metabolites with high mean and low standard deviation will tend to be higher than that of metabolites with low mean and standard deviation. Such a result is obviously not what we want, and only those with large differences between groups should be of high importance.

Line 223.- With which database were the data obtained from HPLC/Masses compared?

Response: The databases used for non-target metabolism search are mzcloud, mzvault, and Masslist. The identified metabolites were annotated using KEGG, HMDB and LIPIDMaps databases.

Line 283. What does Con, Mod y NF means?

Response: Con means the control group, which treated with oral water.

Mod means the model group, which treated with CUMS and oral water.

NF means CUMS rats treated with jasmine tea scented by new process.

Line 307. There are many figures or graphs, this type of results can be summarized with another type of data analysis, for example, non-parametric analysis such as principal components or multivariate.

Response: Thank you for your valuable suggestion. In this part, we just want to indicate that jasmine tea administration increased the neurotransmitters and inflammation among different tissues. The figure 3 can show the differences of biochemical indexes among different tissues.

Line 335- 360. They describe graphs A-D but do not mention whether or not there is a significant difference in the analyzes carried out.

Response: Thank you for your valuable suggestion. We had mentioned the significant difference in the analysis. For example, “The gut microbiota community analysis showed that the species diversity indices (Shannon and Simpson) and species richness indices (chao1 and observed) in Mod was significantly lower than those of the Con, while it could be improved after jasmine tea intervention (Figure 5A)”.

Line 433.- Please explain the meaning of “*” in the heatmap, some * cannot be seen, improve color contrast.

Response: On the heatmap, the X-axis is neurotransmitters and inflammation factors, and the Y-axis is genus. R-values (rank correlation) and p-values are obtained by calculation. R-values are shown in different colors in the figure, and if the P-value is less than 0.05, it is marked with *. The legend on the right is the color interval of different R-values. Meanwhile, the color bar on the left indicates the phylum classification to which the species belongs. * 0.01≤ P <0.05, ** 0.001≤P < 0.01, *** P < 0.001. We also changed the color contrast in the heatmap according to your suggestion.

Line 444-447.- The authors said there are not changes in the presence of metabolites, please put the statistical analysis

Response: we are so sorry that we did not understand this question. In our manuscript, Line 444-447 described that “The PLS-DA method was used to observe the separation among the three groups, and it was found that the metabolites associated with CUMS’s rat and CUMS’s rat companied with jasmine tea there were changed (Figure 9A-C). To further distinguish these specific variations, we used a supervised multivariate OPLS-DA model to clarify the different variables.” We did not say there are not changes in the presence of metabolites.

Line 549-558. Please discuss the results obtained with other authors. If what they got is correct or similar with some other type of food.

Response: Hui He et al. had found that gut microbiota from stressed animals can induce microglial priming in the dentate gyrus, which is associated with a hyper-immune response to stress and impaired hippocampal neurogenesis. Namkwon Kim also observed that FMT from 5xFAD mice into normal C57BL/6 mice (5xFAD-FMT) decreased adult hippocampal neurogenesis and brain-derived neurotrophic factor expression and increased p21 expression, resulting in memory impairment. Microglia in the hippocampus of the 5xFAD-FMT mice were activated, which caused the elevation of pro-inflammatory cytokines (tumor necrosis factor-α and interleukin-1β).

Line 564-566.- The text said “The high level of diversity and richness of gut microbiota is considered as the healthy factor of the host”, but specifically which bacteria are related to the health of the host. Mention related bacteria.

Response: There is growing evidence that the diversity and richness of the gut are important factors affecting host health (Gut-microbiome-expressed 3b-hydroxysteroid dehydrogenase degrades estradiol and is linked to depression in premenopausal females; Oral microbiota dysbiosis alters chronic restraint stress-induced depression-like behaviors by modulating host metabolism, etc). Human gut bacteria Gordonibacter pamelaeae and Eggerthella lenta convert abundant biliary corticoids into progestins through 21-dehydroxylation, thereby transforming a class of immuno- and metabo-regulatory steroids into a class of sex hormones and neurosteroids. It also includes a smaller percentage of Verrucomicrobia, Actinobacteria, Proteobacteria, Bifidobacterium longum and Roseburia intestinalis played a positively role in human health. CUMS altered the gut microbiome, leading to higher relative abundance of some bacteria (Helicobacter, Bacteroides, and Desulfovibrio) and lower relative abundance of some bacteria (Lactobacillus, Bifidobacterium, and Akkermansia). Another research had found that the intervention of Lactobacillus reuteri could suppressed the expression of IL-6, and increased the concentration of BDNF/serotonin, resulting in alleviate gut microbiota-involved depression with colitis in vivo.

Line 656.- Please explain why you compare only with theophylline and theobromine.

Response: The theophylline and theobromine were detected in serum metabolism in this study, both of which are also present in jasmine tea. We speculated that theophylline and theobromine may be related to jasmine tea’s alleviation of depression. Thus, we compare with theophylline and theobromine in this study.

We tried our best to improve the manuscript and made some changes in the manuscript. The changes will not influence the content and framework of the paper. And here we did not list the changes but marked in red in revised paper.

We appreciate for reviewer’s warm work earnestly, and hope that the correction will meet with approval.

We will submit the revised draft now. In order to make the revision complied with the publication requirements of your journal, we hope you can provide valuable comments, we will make further revisions until the paper is accepted.

Thank you for your consideration. We look forward to hearing from you.

Sincerely,

[Yangbo Zhang]

[Shaoyang University, Shaoyang, Hunan]

[13272032875]

[+86-0739-5308282]

[[email protected]]

Reviewer 2 Report

Comments and Suggestions for Authors

The topic is interesting and has an important novelity, however I consider the paper needs to review in order to enhance quality of it and reduce many mistakes identified in the the text.

Abstract needs to contain more specific results instead of generalities. I recommend to review it,

Introduction. It is very general and specific information related with research needs to cover with references and more detail.

Material and Methods

The animal design is not clear and needs to give more information with detial of specific conditions, for example the mechanism used ofr ingest of jasmine..

The hystological analysis should be explain with more detail overall marked on the images the specific evidences identified or the lack of it.

Statistical analysis is covered only for anova but PCA or metagenomic studies are nott covered in this section .

Results and discussion is well prepared and explained.

Author Response

Response to Reviewer Comments

Dear Editors and Reviewers:

Thank you very much for your letter and for the reviewers’ comments concerning our manuscript entitled “The intestinal microbiome combined with metabolomics to explore the mechanism of jasmine tea improving depression in CUMS rats”. Those comments are all valuable and very helpful for revising and improving our paper. We have seriously taken the comments into consideration in revising our manuscript. Revised portion are marked in red in our manuscript. The main corrections in the paper and the responds to comments are as follows:

Reviewer 2

The topic is interesting and has an important novelity, however I consider the paper needs to review in order to enhance quality of it and reduce many mistakes identified in the text.

Abstract needs to contain more specific results instead of generalities. I recommend to review it.

Response: Thank you for your suggestion. We had added more specific results in the abstract based on valuable suggestion (Line 26-29).

Introduction. It is very general and specific information related with research needs to cover with references and more detail.

Response: Thank you for your suggestion. The general and specific information related with research were listed in the introduction.

Material and Methods

The animal design is not clear and needs to give more information with detial of specific conditions, for example the mechanism used of r ingest of jasmine.

Response: Thank you for your advice. The choose of jasmine tea dose in this experiment followed by our previous study (Jasmine Tea Attenuates Chronic Unpredictable Mild Stress-induced Depressive-Like Behavior in Rat Via the Gut-Brain Axis, which was published on nutrients). Jasmine tea soup was prepared with drinking water and taken orally once a day at a concentration of 64.8 mg/kg (1 mL/kg/day) until the end of the experiment. Therefore, in this study, we only selected medium-dose of jasmine tea for related research.

The hystological analysis should be explain with more detail overall marked on the images the specific evidences identified or the lack of it.

Response: Thank you for your valuable suggestion. We had modified the histological analysis based on your suggestion, and marked the specific evidences on the images.

Statistical analysis is covered only for anova but PCA or metagenomic studies are not covered in this section.

Response: Thank you for your valuable suggestion. The statistical analysis was only used to analyze the continuous variable such as sucrose preference, the numbers in OFT, immobility time in FST, and body weight, etc. The statistical analysis of 16S rRNA and serum metabolomics were listed at 2.9.2 (Line 183) and 2.11 (Line 225), respectively.

Results and discussion are well prepared and explained.

Response: Thank you for your affirmation.

We tried our best to improve the manuscript and made some changes in the manuscript. The changes will not influence the content and framework of the paper. And here we did not list the changes but marked in red in revised paper.

We appreciate for reviewer’s warm work earnestly, and hope that the correction will meet with approval.

We will submit the revised draft now. In order to make the revision complied with the publication requirements of your journal, we hope you can provide valuable comments, we will make further revisions until the paper is accepted.

Thank you for your consideration. We look forward to hearing from you.

Sincerely,

[Yangbo Zhang]

[Shaoyang University, Shaoyang, Hunan]

[13272032875]

[+86-0739-5308282]

[[email protected]]
